# A Prospective Clinical Study Evaluating the Efficacy of Intra-Ligamentary Anesthetic Solutions in Mandibular Molars Diagnosed as Symptomatic Irreversible Pulpitis with Symptomatic Apical Periodontitis

**DOI:** 10.3390/healthcare10081389

**Published:** 2022-07-26

**Authors:** Khalid Gufran, Mubashir Baig Mirza, Ali Robaian, Abdullah Saad Alqahtani, Nasser Raqe Alqhtani, Mohammed Alasqah, Abdulaziz Mohammad Alsakr

**Affiliations:** 1Department of Preventive Dental Sciences, College of Dentistry, Prince Sattam Bin Abdulaziz University, Alkharj 11942, Saudi Arabia; ab.alkahtani@psau.edu.sa (A.S.A.); m.alasqah@psau.edu.sa (M.A.); a.alsakr@psau.edu.sa (A.M.A.); 2Department of Conservative Dental Sciences, College of Dentistry, Prince Sattam Bin Abdulaziz University, Alkharj 11942, Saudi Arabia; ali.alqahtani@psau.edu.sa; 3Department of Oral and Maxillofacial Surgery and Diagnostic Science, College of Dentistry, Prince Sattam Bin Abdulaziz University, Alkharj 11942, Saudi Arabia; n.alqhtani@psau.edu.sa

**Keywords:** anesthesia, intra-ligamentary, injection, mandible, pulpitis, periodontitis, symptomatic

## Abstract

Accomplishing painless endodontic treatment, especially in the mandibular molar region, is challenging. Hence, the aim of the study was to compare the efficacy of 2% lidocaine and 4% articaine when used as supplemental intra-ligamentary (IL) anesthesia in mandibular molars having symptomatic irreversible pulpitis with symptomatic apical periodontitis after failed Inferior Alveolar Nerve Block (IANB) injection. In this prospective study, one-hundred and forty-seven adult patients diagnosed with irreversible pulpitis in a mandibular tooth were included who received IANB with 1.8 mL of 2% lidocaine with 1:100,000 epinephrine. Patients who experienced pain were recorded using the Heft–Parker visual analog scale (HP-VAS score ≥ 55 mm) and received supplement intra-ligament injection with either4% articaine or 2% lidocaine with 1:100,000 epinephrine. Supplementary intra-ligament injections resulted in 82.6% and 91.3% of profound anesthesia in the first molar region for 2% lidocaine and 4% articaine, respectively. Similarly, an additional IL injection of articaine success percent (78.9%) in the second molar region was higher than lidocaine (63.1%). The overall success ratio revealed no significant difference in achieving profound anesthesia of either solution. In this study population, there was no difference in the success ratio of anesthesia between 2% lidocaine and 4% articaine when used as supplemental IL injection.

## 1. Introduction

Every patient has the right to expect comfort during any treatment procedure. In dentistry, pain management is challenging due to the complex neurological innervation of craniofacial structure. Furthermore, its uniqueness depends on the procedures and site of involvement. Achieving profound anesthesia for a tooth diagnosed having synergistic symptomatic irreversible pulpitis with symptomatic apical periodontitis is an immense parameter in a successful endodontic practice [1]. Scientific evidence supports that accomplishing painless root canal treatment (RCT) is still challenging, especially in the mandibular molar region with a thick cortical bone [1,2].

Numerous anesthetic techniques have been experimented with to achieve profound pain analgesia in mandibular molars. Inferior Alveolar Nerve Block (IANB) is a primary method to obtain anesthesia in the mandibular region due to its safety, reliability, and ease in delivering the target solution [2,3]. However, evidence from a recent meta-analysis revealed that the standalone anesthetic efficacy of IANB ranged from 43% to 83% [2,4,5,6,7]. One of the possible reasons for this could be the failure to locate the lingua [8]. Furthermore, the literature shows several reports of supplementary injection techniques and solutions in various cases for pain management in irreversible pulpitis patients [9,10,11,12,13,14].

Intra-ligamentary injections have been used as a standalone technique to achieve anesthesia in endodontic treatment procedures. A study that used 194 patients to determine pain and complications associated with the administration of the intra-ligamentary injection technique found various benefits such as localized soft tissue anesthesia, decreased pain on injection, and minimal pain during procedure. It was also found useful in patients with severe gag reflex, trismus, etc. [15]. Another recent randomized clinical trial that used 72 patients to assess the effectiveness and complications of intra-ligamentary anesthesia with IANB found both techniques to be equivalent in effectiveness for pain control, and at the same time, intra-ligamentary injections showed significantly less pain during injection [16]. A recent systematic review and meta-analysis that evaluated the success rate of supplementary IL injection in the mandibular teeth of patients with irreversible pulpitis also found that these injections provide better success rates for anesthesia and increased the efficacy of anesthesia [17]. It might be noted that there are few reports of ILA causing some damage to periodontal tissue, bone, and even cause root resorption if not used cautiously. However, all such reported instances were reversible and did not lead to permanent damage [16,18].

Among the available techniques, intraosseous and intra-ligamentary techniques posed a viable method to increase the success rate of failed IANB [19,20]. The need for specialized devices and their hemodynamic effects on the supplementary intraosseous process made practitioners opt for intra-ligamentary methods to perform painless intraoral procedures [9,14]. Recent studies suggest that supplementary intra-ligament (IL) injection and IANB produced more successful anesthesia. However, none of the studies assessed the efficacy of IANB and supplementary IL combination in mandibular molars having symptomatic pulpitis with apical periodontitis.

Most of the commonly used dental anesthetics include 2% lidocaine and 4% articaine [3,21,22,23,24]. Although considerable evidence supports the safety and efficacy, mixed results also prevail, necessitating multicentric studies globally [24,25,26]. None of the anesthetic methods/drugs resulted in obtaining foreseeable and plausible anesthesia for endodontic procedures in symptomatic pulpitis patients, necessitating the need for reliable techniques and drug combinations among dental practitioners [27].

Since the level of anesthesia obtained varied from person to person [14,28], the authors took an interest in comparing the efficacy of 2% lidocaine and 4% articaine when used as supplemental intra-ligamentary anesthesia in mandibular molars having symptomatic irreversible pulpitis with symptomatic apical periodontitis. The null hypothesis of the study is that there is no difference in the efficacy of these supplementary intra-ligament injections among patients having symptomatic irreversible pulpitis with apical periodontitis after failed IANB injection in mandibular molars.

## 2. Materials and Methods

### 2.1. Ethical Approval

This prospective study was conducted at the Specialist clinics, College of Dentistry, Prince Sattam bin Abdulaziz University Alkharj (PSAU), Saudi Arabia. Ethical approval was obtained from the University Institutional Review Board, PSAU (SCBR-016-2022), following which the study was carried out following the ethical principles for medical research on humans established by the Helsinki protocol (version 17c, 2004).

### 2.2. Patient Selection

One hundred and forty-seven adult patients who attended the specialist clinic, College of Dentistry, were invited to participate in the study from 10 January to 10 April 2022. Eligibility criteria included all participants who had vital mandibular first and second molars with a finding of symptomatic irreversible pulpitis with symptomatic apical periodontitis based on the classification given by the American Association of Endodontics.

However, patients below 18 years of age, patients with known allergies to anesthetic solutions, patients with medical history other than American Society of Anesthesiologists (ASA)-I, any other pulpal and periapical diagnosis other than symptomatic irreversible pulpitis with symptomatic apical periodontitis, and inability to give informed consent were excluded from the study.

### 2.3. Intervention

A clinician not involved in giving anesthesia confirmed objective pulpal diagnosis using the affirmative reaction to electric pulp test (Digitest^®^ Parkell, Edgewood, NY, USA) and a response grade of enduring pain to cold test using Roeko Endo-Frost (Roeko, Raiffeisenstrabe 30, Langenau, Germany). “Heft–Parker Visual Analogue Scale” (HP-VAS) was employed to rank the patient pain and discomfort tendency. The HP-VAS is a 170 mm line split into domains, each describing a certain level of pain [14,29]. After which, based on percussion, palpation, and bite test, a periapical diagnosis was obtained clinically.” Whether the tooth can be restored or not?” was assessed synergistically with bitewing radiographs and clinical findings. If the diagnosed tooth could be restored and isolated with a rubber dam, it was considered eligible.

### 2.4. Anesthesia Protocol

Initially, an experienced clinician (Principal investigator) explained the treatment procedures and pain scales to the patients. The teeth were then anesthetized and treated by the same specialist dentist (Principal investigator). These patients received primary block anesthesia employing one carpule of approximately 1.8 mL of 2% lidocaine containing epinephrine in 1:100,000 concentration (Octocaine 100, Novocol Pharma, Cambridge, ON, Canada) after the injection area was dried using sterile gauze. Additionally, a clean cotton tip applicator applied topical anesthesia of 20% benzocaine (Prime gel, Chicago, IL, USA) for 1 min prior to the block anesthesia [5,9].

The anesthetic solution was injected with a disposable syringe measuring 0.4 × 30-mm/27-G needle (C-K ject, C-K Dental Co. Ltd., Bucheon, Korea). After palpating the mandibular ramus’s anterior border, each patient’s coronoid notch was located. The target area was then aspirated, and care was taken to deposit anesthetic solution at a slow rate to avoid pain. After waiting for approximately 10 min, the principal investigator inquired about the lip numbness of the patient. If the participant reported an affirmative response to regional anesthesia, teeth were isolated using a rubber dam and an access opening commenced.

Patients were asked to raise their hands in case of pain during the procedure and rated it on HP-VAS. If the scored pain was more than 54 score on HP-VAS, the rubber dam was removed, and a supplemental ligament injection (2% lidocaine or 4% articaine) was planned to complete the procedure comfortably.

The injection site was cleaned, and a 27 G 0.4 × 21 mm needle was inserted at an angle of 30° to the long axis of the tooth, initially at the mesiobuccal aspect and later in the distobuccal parts of each root. After reaching the target area between the tooth and crestal bone, 0.2 mL of the anesthetic solution was injected into the periodontal ligament under strong back pressure on each side. A coldtest was performed to identify the presence of anesthetic failure. HP-VAS was used to record pain after the cold testing. When there was no response to cold, the tooth would be isolated and an access cavity prepared.

If the patient experienced pain and discomfort during the therapy, the clinician would give an additional shot of IL injection, and the case was considered a ‘failure’. The operator declared the ‘success’ of IANB supplemental with IL injection if he/she could access and instrument the tooth without pain or mild pain (VAS < 54 mm).

### 2.5. Data Analysis

Data were analyzed for descriptive statistics using SPSS for Windows, version 21 (IBM, SPSS, Chicago, IL, USA).

## 3. Results

Eighty-two patients participated in this study. Among 147 IANB injections, “successful anesthesia” was reported in less than 50% of teeth, i.e., 1st molars (36/82) and 2nd molars (27/62). According to the location in the mandible, the percent of failed IANB injections were 56.09 and 58.46 for first and second mandibular molars, respectively. Further, analyzing the output of IANB gender-wise revealed that more females were unable to achieve anesthesia 1st molars (17/43) and 2nd molars (11/35) than males 1st molars (19/39) and 2nd molars (15/30) with IANB alone. The gender-wise distribution of failed primary INAB anesthesia in symptomatic irreversible pulpitis patients is shown in Table 1.

Failed IANB, followed by supplementary intra-ligament injections, resulted in 82.6% and 91.3% of profound anesthesia in the first molar region for 2%lidocaine and 4% articaine, respectively. Similarly, an additional IL injection of articaine success percent (78.9%) in the second molar region was higher than lidocaine (63.1%). However, the overall success ratio revealed no significant difference in achieving profound anesthesia of either solution.The success ratios of supplemental intra-ligamentary lignocaine and articaine anesthesia in symptomatic irreversible pulpitis patients are shown in Table 2. Here, success is the percentage of successful anesthesia for lignocaine and articaine separately, whereas the success ratio is the difference between success percent of lignocaine to that of articaine. Example—for 1st molar, the success ratio is 19/23:21/23 = 0.9.

## 4. Discussion

Since the success of an endodontic procedure at any stage depends on profound anesthesia, achieving good anesthesia is essential for controlling pain and alleviating fear and anxiety [30,31]. The present study showed that “IANB with supplemental intra-ligamentary infiltration” is a successful alternative to “IANB alone” in achieving anesthesia in patients having symptomatic irreversible pulpitis with apical periodontitis in the mandibular molar region. The most widely employed IANB in achieving local anesthesia for teeth in the lower jaw reported higher failure rates [9,10,32,33]. Generally, the achievement of soft tissue anesthesia in IANB remains high, but when vital inflamed pulp tissues are encountered as in cases of symptomatic irreversible pulpitis, the success rates usually are on a lower end. In irreversible pulpitis, the failure of IANB was estimated to range between 43% and 83% [2,5,32]. IANB has a high failure rate, and success rates are even lower when applied for the treatment of mandibular posterior teeth with irreversible pulpitis [32,34]. This led to the search for an efficient anesthetic technique/solution over the decades that will achieve efficient anesthesia. Additionally, in symptomatic irreversible pulpitis cases, IANB showed less than a 30% success rate of anesthesia [4,5,17] due to the inflammation-related activation of specific receptors such as tetrodotoxin-resistant receptors [35] directly resisting the concentration of local anesthetic agents and reducing their efficacy [33,36,37].

The supplemental intra-ligament injection is a viable complement to failed IANB to secure anesthesia when the dentist wants anesthesia for a little more time and the patient can no more bear lip and tongue numbness [9,33,38,39]. Studies show that the percent of supplementary IL injection for mandibular molars that successfully anesthetized teeth with irreversible pulpitis varied between 48 and 92% [9,19,39,40,41,42]. In the current study, authors reported the success rate for supplementary intra-ligament injection for symptomatic pulpitis with apical periodontitis patients in the range said earlier. It is interesting to note that although the patient had the highest pain perception, the IANB and supplemental IL combo produced efficient anesthesia in the majority of the patients.

Measuring pain precisely is difficult in patients as it varies based on individual pain perception, intensity, and various sensory/influential factors. It is quite challenging to enumerate and homogenize pain objectively among individuals. Hence, investigators took keen care to validate pulpal anesthetic success using a mixed method of analysis, i.e., EPT, thermal test, and subjective VAS.

Comparing the present combination with other techniques/solutions earlier resulted in the following conclusions. IANB and supplemental buccal infiltration reported a success rate of 42–88% [5,11,43,44,45,46]. Dianat (2020) carried out a study of a different combination of primary IANB and supplementary buccal and inter-septal injection revealed a success rate of 80% anesthesia [27]. In our study, supplementary IL anesthesia produced much higher anesthesia (82.1% and 91.3%) than other techniques [27]. However, comparing the present study results with the previous study should be proceeded with caution due to variation in the location of the posterior tooth, the method used in pulpal diagnosis (asymptomatic or symptomatic pulpitis), and the method employed to assess the anesthetic success rate. The strength of the present study is the utilization of both subjective and objective methods for pain measurement. Selection/reporting bias is taken care of by allocation concealment. Furthermore, clinicians not involved in the study assessed the level of anesthesia before and during the treatment procedure. However, there are some limitations, too. The electric pulp test could also have been used to test the anesthesia. More groups could have been included in the study to test other than the use of VAS for assessing the pain threshold. Future recommendations include randomized clinical trials and prospective studies on different techniques/anesthetics in patients having symptomatic pulpitis with apical periodontitis.

## 5. Conclusions

In patients having symptomatic irreversible pulpitis with apical periodontitis in the mandibular molar region, “IANB with supplemental intra-ligamentary infiltration” is a successful alternative to “IANB alone” technique in achieving anesthesia. Within the study’s limits, there is no difference in the success ratio of anesthesia between 2% lidocaine and 4% articaine when used as supplemental IL injection. However, none of the tested methods or solutions ensured profound anesthesia in all cases.

## Figures and Tables

**Table 1 healthcare-10-01389-t001:** Gender-wise distribution of failed primary inferior alveolar nerve block anesthesia in symptomatic irreversible pulpitis patients.

Area Anesthetized	Gender	Total	Success	Failure	Percent of Failed Injections
	Total	82	36/82	46/82	56.09
**1st Molar**	Males	39	19/39	20/39	51.28
	Females	43	17/43	26/43	60.46
	Total	65	27/65	38/65	58.46
**2nd Molar**	Males	30	15/30	15/30	50.00
	Females	35	11/35	23/35	65.71

**Table 2 healthcare-10-01389-t002:** Thesuccess ratio of supplemental intra-ligamentary lignocaine and articaine anesthesia in symptomatic irreversible pulpitis patients.

Area Anesthetized	Anesthesia	Total	Success (% of Successful IL Injections)	Failure (% of Failed IL Injections)	Success Ratio
1st Molar	IANB	82	36/82 (43.9)	46/82 (56.0)	0.9
	IANB/Lignocaine	23	19/23 (82.6)	4/23 (17.3)	
	IANB/Articaine	23	21/23 (91.3)	2/23 (8.6)	
2nd Molar	IANB	65	27/67 (40.2)	38/67 (56.7)	0.8
	IANB/Lignocaine	19	12/19 (63.1)	7/19 (36.8)	
	IANB/Articaine	19	15/19 (78.9)	4/19 (21.0)	

## Data Availability

The data used to support the findings of this study are available from the corresponding author upon request.

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
