# Peer review of "A Prospective Clinical Study Evaluating the Efficacy of Intra-Ligamentary Anesthetic Solutions in Mandibular Molars Diagnosed as Symptomatic Irreversible Pulpitis with Symptomatic Apical Periodontitis"

_healthcare, 2022, doi:10.3390/healthcare10081389_

Round 1

Reviewer 1 Report

Dear authors!

Irreversable pulpitis are our everyday stress if the IANB works unproperly. And it is a very good decision to add PDL injection to the clinical protocol. I have a list of questions which i hope will find answers.

1. Was the concentration of the epinephrine the same for 2% lidocaine and 4% articaine?

2. When we use a local anesthetic with vasoconstrictors and, in principle, when working with patients who have pain, it is important to remember the assessment of the functional state. Have you measured your blood pressure and pulse? What are the inclusion and exclusion criteria from the study?

3. "About five milliliters of anesthetic solution were injected with a disposable syringe measuring 0.4x30-mm/27-G needle"

Why do you use 5.0 ml of local anesthetic? Do you know about optimal dosage and volume of the injectable space?

4. 'A coldtest was performed to check the level of anesthesia'

How do you measure the level of the anesthesia in this case? Such procedure is not this procedure is not precise, since the pain threshold in all patients is different

Also some questions are about results.

1. Please provide a table with direct information about successful IANB without IL and its combination. 

2. "Among 147 IANB injections, "successful anesthesia" was reported in less than 50% of teeth"

It is a very low result, seems to me you have to change your IANB technique

3. It is nothig said about EPT measurment in the results, also nothing about procedure of use: after 1, 5, 30 or 60 from the injection. This aspect is the only objective for pain management measurement.

4. You have to perform more groups if you use differect local anesthetics.

2% lidocaine for IANB, IANB+IL

4% articaine for IANB, IANB+IL

Discussion and conclusion.

We all know about effectiveness of the buccal injection as an edition to the IANB. Why didn't you focus your attention at this aspect?

In conclusion you write "Within the study's limits, there is no difference in the success ratio of anesthesia between 2% lidocaine and 4% articaine when used as supplemental IL injection"

For sure after IANB there is no difference in general but your research is not really evident without without the use of precise methods for assessing the pain threshold. It is not enough to use the VAS for such a conclusion, especially if we take into account the pharmacological activity of lidocaine, which is inferior to articaine, and the concentration of the vasoconstrictor.

Finally, you need to review the bibliography and update it. Most references are 10 or more years old, while the effectiveness of local anesthesia is constantly discussed and the results are published.

Reviewer 2 Report

It is an interesting study, but need more evidence to make the point valid

In order to improve the manuscript, it need, structuring and summarising the results, and well-placed and discussed outcomes of study.

How did you select the sample number and how long the study was conducted?

Why only molar was chosen?

Expand the discussion to discuss the results and outcomes and future perspectives

Add in the limitation and future recommendation for your study in clinical practicality based on your results.

Reviewer 3 Report

This study evaluated the efficacy of 2% lidocaine and 4% articaine using in supplemental intra-ligamentary (IL) anesthesia in mandibular molars having symptomatic irreversible pulpitis with symptomatic apical periodontitis after failed Inferior Alveolar Nerve Block (IANB) injection. The manuscript is clearly written and the methodology is appropriate. I have several concerns:

1. In table 2, please explain what’s the difference between column “Success (% of successful II injections)” and “Success Ratio”.

2. It would be better to list the average VAS number with SD. Then people may have more sense about the level of the pain.

3. Several studies already analyzed the efficiency of 2% lidocaine and 4% articaine using in IL anesthesia, the author need point out the novelty of this study.

4. In “Anesthesia protocol” part, at the beginning of paragraph 2 “About five milliliters of …” should be a typing mistake, might be “five microliters of …”.

Round 2

Reviewer 1 Report

Dear authors, thank you for your responce. 

However, unfortunately your comments force me to ask a few more questions, as there is still no clarity. There is enough information in the literature that the combination of block and PDL with 4% articaine gives a tangible advantage over other techniques and preparations. In many ways, this is the advantage of the pharmacological characteristics of the drug.

1. Dosage correction relative to the block is understandable, but there is no information about the dosage on PDL which you have used.  

2. We know that the health of the periodontal tissues is a necessary condition for the performance of the PDL, but you did not say anything about this.

3. You only confirm my words with your quotes "low success rates of IANB are valid" - this problem has long been known and there are a sufficient number of methods to solve it. The most effective way is premedication, such as NsAID. If you have an ASA-1 patient, why did you not use this category of drugs at the initial stage?

4. Finally, the main question - if we know in advance that the block is low effective for this category, then why did you choose it anyway, and not a high blockade like Gow-Gates?

5.My comment on the list of references is still relevant: for a long time the majority is more than 10 years old, which makes it difficult to conclude about the relevance of the study. If you refer to basic materials to support your scientific opinion, then a logical question is - what is the novelty of your research, besides confirming the obvious data?

Reviewer 2 Report

The authors have improved the manuscript.

Author Response

Dear Reviewer,

I would like to thank you once again for being patient and considerate with us regarding the manuscript.

  1. Moderate English changes required?

Response: The manuscript has been revised to include moderate language corrections. 

Round 3

Reviewer 1 Report

Dear authors!

Thank yu for your answers. I am satisfied

Author Response

Dear Reviewer,     

Thank you for the kind comments.